# Delineating *Mycobacterium abscessus* population structure and transmission employing high-resolution core genome multilocus sequence typing

Margo Diricks[1,2,3] ✉, Matthias Merker[2,4], Nils Wetzstein[5], Thomas A. Kohl [1,2,3], Stefan Niemann[1,2,3,7] & Florian P. Maurer [2,3,6,7]

*Mycobacterium abscessus* is an emerging multidrug-resistant non-tuberculous mycobacterium that causes a wide spectrum of infections and has caused several local outbreaks worldwide. To facilitate standardized prospective molecular surveillance, we established a novel core genome multilocus sequence typing (cgMLST) scheme. Whole genome sequencing data of 1991 isolates were employed to validate the scheme, re-analyze global population structure and set genetic distance thresholds for cluster detection and taxonomic identification. We confirmed and amended the nomenclature of the main dominant circulating clones and found that these also correlate well with traditional 7-loci MLST. Dominant circulating clones could be linked to a corresponding reference genome with less than 250 alleles while 99% of pairwise comparisons between epidemiologically linked isolates were below 25 alleles and 90% below 10 alleles. These thresholds can be used to guide further epidemiological investigations. Overall, the scheme will help to unravel the apparent global spread of certain clonal complexes and as yet undiscovered transmission routes.

Non-tuberculous mycobacteria (NTM) comprise all *Mycobacterium* species that do not cause tuberculosis or leprosy[1]. NTM bacteria are ubiquitous in natural aquatic and soil environments, but have also been frequently isolated from tab water, swimming pools and showers[2–5]. Although bacteria of most NTM species are harmless for humans, some are (opportunistic) pathogens that can cause severe disease especially in immunocompromised patients[6].

Of particular clinical relevance is the rapidly growing NTM *M. abscessus* (Mab), which is increasingly being reported as the etiological agent of severe respiratory, skin and mucosal infections and is extremely difficult to treat due to intrinsic resistance against several antibiotics[6–9].

Pulmonary infections with Mab are especially prevalent among individuals with pre-existing structural lung diseases such as cystic fibrosis (CF) and bronchiectasis. In these patients, Mab infection leads to an accelerated decline in lung function and is associated with increased mortality[10,11]. It is thought that CF patients predominately acquire NTM infections from the environment, although there have

[1]Molecular and Experimental Mycobacteriology, Research Center Borstel, Borstel, Germany. [2]German Center for Infection Research (DZIF), partner site Hamburg-Lübeck-Borstel-Riems, Borstel, Germany. [3]National and WHO Supranational Reference Laboratory for Mycobacteria, Research Center Borstel, Leibniz Lung Center, Borstel, Germany. [4]Evolution of the Resistome, Research Center Borstel, Borstel, Germany. [5]Department of Internal Medicine, Infectious Diseases, University Hospital Frankfurt, Goethe University, Frankfurt am Main, Germany. [6]Institute of Medical Microbiology, Virology and hospital Hygiene, University Medical Center Hamburg-Eppendorf, Hamburg, Germany. [7]These authors contributed equally: Stefan Niemann, Florian P. Maurer. ✉e-mail: mdiricks@fz-borstel.de

been reports of possible indirect person-to-person transmission within healthcare facilities through fomites or long-living infectious aerosols[2,12-15].

Mab can also cause extra-pulmonary disease such as skin and soft tissue infection (SSTI) in healthy persons and sporadically otitis media (ear infections) in children[16]. Many SSTI outbreaks affecting multiple individuals have been related to surgical or cosmetic procedures likely due to contaminated water sources or medical devices, e.g. hospital water supply, wading pool, tattoo ink, ultrasonography gel, and bronchoscopes[17-23].

Based on multiple genomic comparison studies, Mab strains have been divided into three subspecies: *M. abscessus* subsp. *abscessus* (Mab$_A$), *M. abscessus* subsp. *bolletii* (Mab$_B$), and *M. abscessus* subsp. *massiliense* (Mab$_M$)[24]. Recently, several clusters of closely related isolates within both Mab$_A$ and Mab$_M$ have been identified. These so called dominant circulating clones (DCC) have been isolated from both CF and non-CF patients across the whole globe, are thought to have emerged around 1960[12,25,26] and have been associated with increased virulence, higher rates of resistance, and worse clinical outcomes compared to unclustered isolates[12].

Previous molecular outbreak investigations, source tracking, and population structure analyses of Mab have mainly been performed by repetitive sequence-PCR (rep-PCR), random amplified polymorphic DNA typing (RAPD PCR), multilocus sequence typing (MLST), pulsed-field gel electrophoresis (PFGE) and core genome single nucleotide polymorphism (cgSNP) analysis[4,27-29]. While cgSNP clearly offers the highest resolution for outbreak investigations, the lack of standardized bioinformatic pipelines and nomenclature scheme limits the application to individual retrospective investigations[30-32].

For prospective molecular surveillance, core genome multilocus sequence typing (cgMLST) has been shown to provide easy inter-laboratory comparability and a continuous comparative analysis, which facilitates real-time multicenter outbreak investigations for different pathogens such as *Pseudomonas aeruginosa*[33], *Mycobacterium tuberculosis*[34], *Listeria monocytogenes* and *Paenibacillus larvae*[30,33-35]. The gene-by-gene comparison approach of cgMLST relies on a fixed set of conserved genes distributed across the entire genome that are present in the majority of strains within a species. An inherently standardized and expandable nomenclature implementation translates the DNA sequences of the respective genes into integer allele numbers, which can then easily be compared between strains[36].

In this study, we developed a novel cgMLST scheme using 97 diverse genomes, to allow harmonized whole genome sequencing based typing of all three Mab subspecies. Then, we evaluated its potential for population structure analysis, outbreak investigations and transmission analysis as well as compatibility with traditional 7-loci MLST and the cgSNP-based approach by utilizing a large set of 1991 isolates including, among others, dominant circulating clones, longitudinal pulmonary isolates from chronically infected cystic fibrosis (CF) patients and isolates from three extra-pulmonary Mab outbreaks.

## Results

### Design and technical validation of a stable Mab cgMLST scheme
We used Mab$_A$ type strain ATCC19977 and 96 additional publicly available assemblies (scheme creation set) from a genetically diverse and global set of Mab isolates to define a hard core genome with SeqSphere$^+$ software (Supplementary Data 1, Supplementary Figs. 1-3 and Supplementary Methods 1). The scheme creation set included representatives for all subspecies, for seven previously defined DCCs[26] as well as non-DCC strains, which were collected in at least 11 different countries. The resulting cgMLST scheme consists of 2904 loci (Supplementary Data 2), representing 59% of the gene set from Mab$_A$ type strain ATCC19977. To assess the robustness of the scheme, we compared cgMLST profiles obtained for draft genomes generated from the same sequencing read set with different assembly approaches for 30

diverse isolates (Technical validation set; Supplementary Data 3 and 4). Details are available in Supplementary Methods 2. In summary, cgMLST analysis was very fast and cgMLST profiles (i.e. allele numbers) were identical for draft genomes generated with different assembly pipelines (i.e. SeqSphere+[37] and shovill[38]) using different assemblers (i.e. skesa[39] and SPAdes[40]) and different read pre-processing steps (i.e. performing default trimming and/or read error correction or not). On the other hand, larger differences up to 19 distinct alleles between cgMLST profiles were observed when de novo assembly approaches were compared with a mapping approach (Supplementary Fig. 4).

### Analysis of the global population structure
To further validate whether the scheme works for Mab strains from different phylogenetic groups and also, classifies Mab strains according to the known global population structure, we performed cgMLST analysis on 1797 isolates, including 1110 strains belonging to Mab$_A$, 563 to Mab$_M$ and 124 to Mab$_B$ (Supplementary Data 1). For 1786 out of 1797 (99.4%) datasets, more than 95.0% good cgMLST targets were found and for 1796 (99.9%) more than 90% of the cgMLST genes were present, indicating a stable core genome applicable for all Mab strains (Supplementary Fig. 1). The strains with less than 95% good cgMLST targets (8 Mab$_M$, 2 Mab$_A$ and 1 Mab$_B$) were removed from further analyses. The neighbor-joining (NJ) tree calculated from pairwise allelic distances of the remaining 1786 isolates revealed that the subspecies classification derived from phylogenetic position in the initial mash distance-based tree (Supplementary Fig. 1) correlated well with cgMLST-based phylogeny (Fig. 1). Consistent with this finding, all isolates belonging to Mab$_A$, Mab$_M$ and Mab$_B$ were most closely related (i.e. had the lowest amount of allele differences) to the type strains for subsp. *abscessus* (strain ATCC19977; accession NC_010397.1), *massiliense* (JCM 15300; NZ_AP014547.1) and *bolletii* (BD; NZ_AP018436.1), respectively. These three type strains (Supplementary Data 5) can thus be used for distance-based classification of new isolates at subspecies level, without the need for phylogenetic tree building.

Strains that were previously classified within a DCC complex based on cgSNP/Fastbaps analysis by Ruis and coworkers[26] also clustered together in the cgMLST-based phylogeny (Fig. 1). The mean intra DCC pairwise genetic distances ranged between 26 (DCC6) and 127 (DCC3) alleles (Supplementary Data 6). The majority of strains within a DCC belonged to one single MLST sequence type (ST). For DCC3, however, two different subclades were identified in the phylogenetic tree that were represented also by two distinct ST types (Fig. 1 and Supplementary Data 6). The differentiation into two groups, i.e. DCC3a (ST33) and DCC3b (ST37), was further supported by the bimodal distribution of the pairwise distances among DCC3 strains (Supplementary Fig. 5).

Next, isolates with unknown DCC status that were positioned within a clade in the NJ tree containing isolates with known DCC status were classified into the corresponding DCC (Fig. 1 and Supplementary Data 1). Using a set of eight representatives, one for each DCC (including 3a, b; Supplementary Data 5), we found a clear separation between intra-DCC pairwise distances (i.e. distances between the representative of the DCC and isolates belonging to the same DCC) and pairwise distances between the representative and isolates not belonging to the corresponding DCC (Supplementary Fig. 6). More concrete, the majority of DCC strains had less than 250 allele differences compared to the corresponding reference genome (Supplementary Fig. 6).

### Transmission and outbreak analysis
To determine an allele threshold that can be used to classify strains in genomic clusters as indicators for possible epidemiologically linked cases, we investigated the genetic diversity within outbreaks and putative transmission clusters, which were previously defined based on cgSNP analysis. In particular, we calculated pairwise allele distances

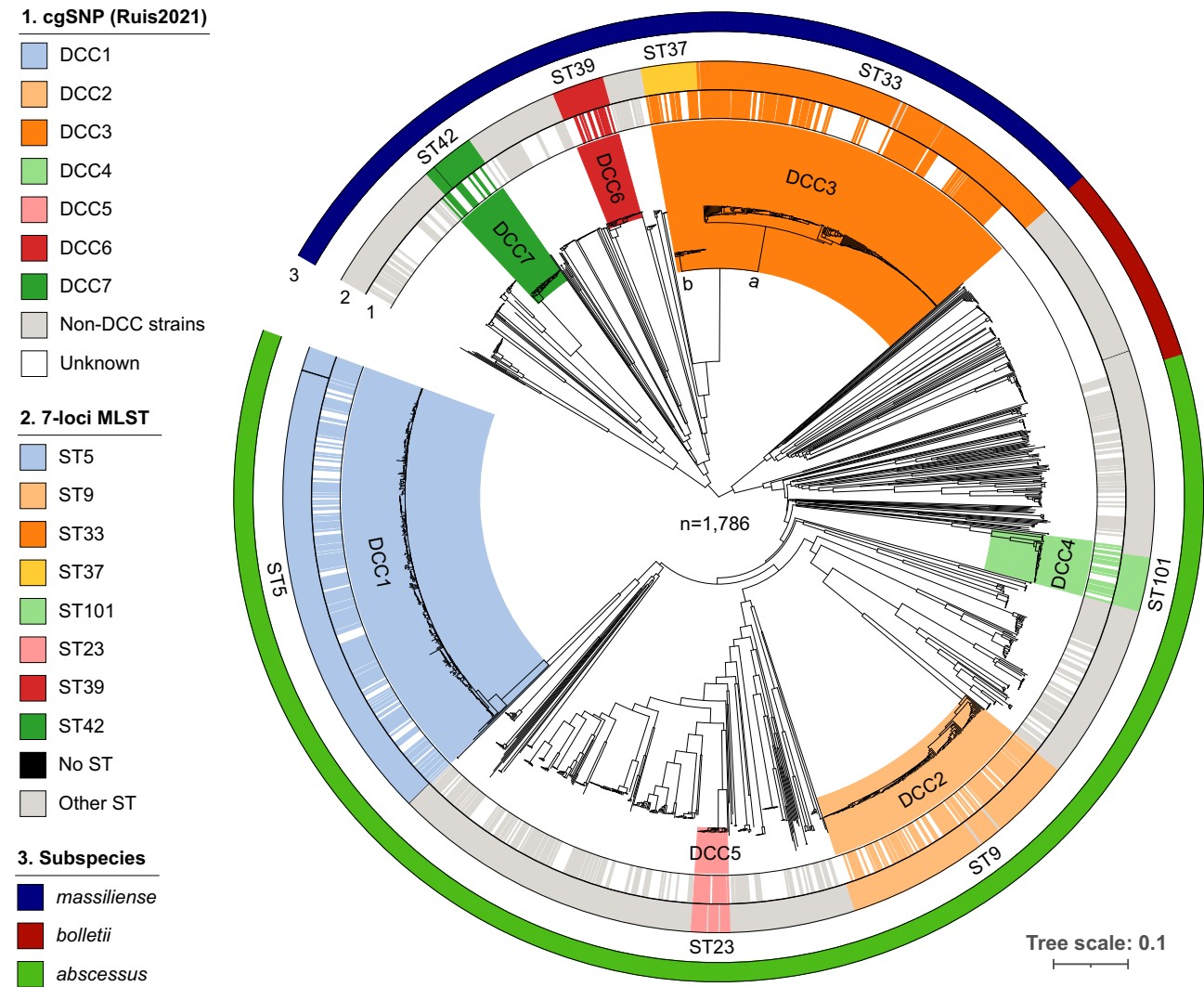

**Fig. 1 | CgMLST-based neighbor-joining tree comprising 1786 *Mycobacterium abscessus* isolates.** 1. Strains that were previously classified into one of seven dominant circulating clones (DCC) based on core genome single nucleotide polymorphism (cgSNP)/FastBAPS analysis[26]. DCC clades (inner circle) are delineated and colored based on the most recent common ancestral node comprising all strains with known DCC classification. 2. Sequence types (ST) according to the nomenclature of the 7-loci MLST scheme hosted on pubMLST[47]. 3. Mash distance-based subspecies classification.

and re-calculated SNP distances between 76 isolates (Supplementary Data 3 and 7) from three extra-pulmonary outbreaks and 12 putative intra-hospital transmission clusters: (i) an outbreak of post-surgical wound infections in Brazil[41], (ii) an outbreak of post-tattooing skin infections in Quebeq[21], (iii) an outbreak of otitis media in paediatric patients in Quebeq[21], and (iv) pulmonary isolates from predominantly CF patients attending clinics in Papworth, Seattle, Milan, Florence or Frankfurt[12,15,42,43]. For some of these CF patients, indirect cross-infection within the hospital was proposed, while for the others, epidemiological investigations did not support this transmission mode (Supplementary Data 7).

Except for two clusters (Frankfurt_CF_C2 [DCC2] and Italy_CF_A1 [DCC1]), all isolates within previously defined clusters were more closely related to each other than to isolates from another cluster (Fig. 2). Interestingly, isolates from an otitis media (OM) outbreak in Quebec were closely related (16 allelic/SNP mismatches) with isolates from a pulmonary Mab outbreak among CF patients in Papworth (Fig. 2).

For most closely related isolates, pairwise SNP and allele distances were in the same range (Fig. 2). However, for two Mab$_A$ isolates from cluster Italy_CF_A4 (GI2 and IR1), the pairwise SNP distance compared to the other cluster members MC1 and TE1 was much

higher (>100 SNPs) compared to the number of allelic differences (<10 alleles). More detailed analysis revealed that the majority of these SNPs in the GI2 and IR1 genome were concentrated in 20 consecutive genes (MAB_1023c-1042c), eight of which were not included in the cgMLST scheme (Supplementary Data 8). BLASTN search with this high-density SNP region revealed a higher total blast score for *Mycobacterium immunogenum* (25,261 with 95% coverage and 87.21% identity) compared to the top Mab hit (24,515 with 89% coverage and 87.37% identity), further pointing towards a putative recombination event.

The pairwise genetic distance between any two isolates belonging to the same extra-pulmonary outbreak was less than 15 alleles with a median of 4 alleles (Fig. 3, group A). Suspected indirect nosocomial transmission within CF centers, on the other hand, was characterized by less than 25 alleles with a median of 2 alleles (Fig. 3, group B). Overall, 95% of pairwise distances between isolates with epi links (i.e. group A and B) were less than 10 alleles. The median pairwise allele distance between clustered isolates with no epidemiological links was significantly higher compared to isolates with supporting evidence for intra-hospital transmission or isolates from well-defined extra-pulmonary outbreaks, but still below 25 alleles (Fig. 3, group C).

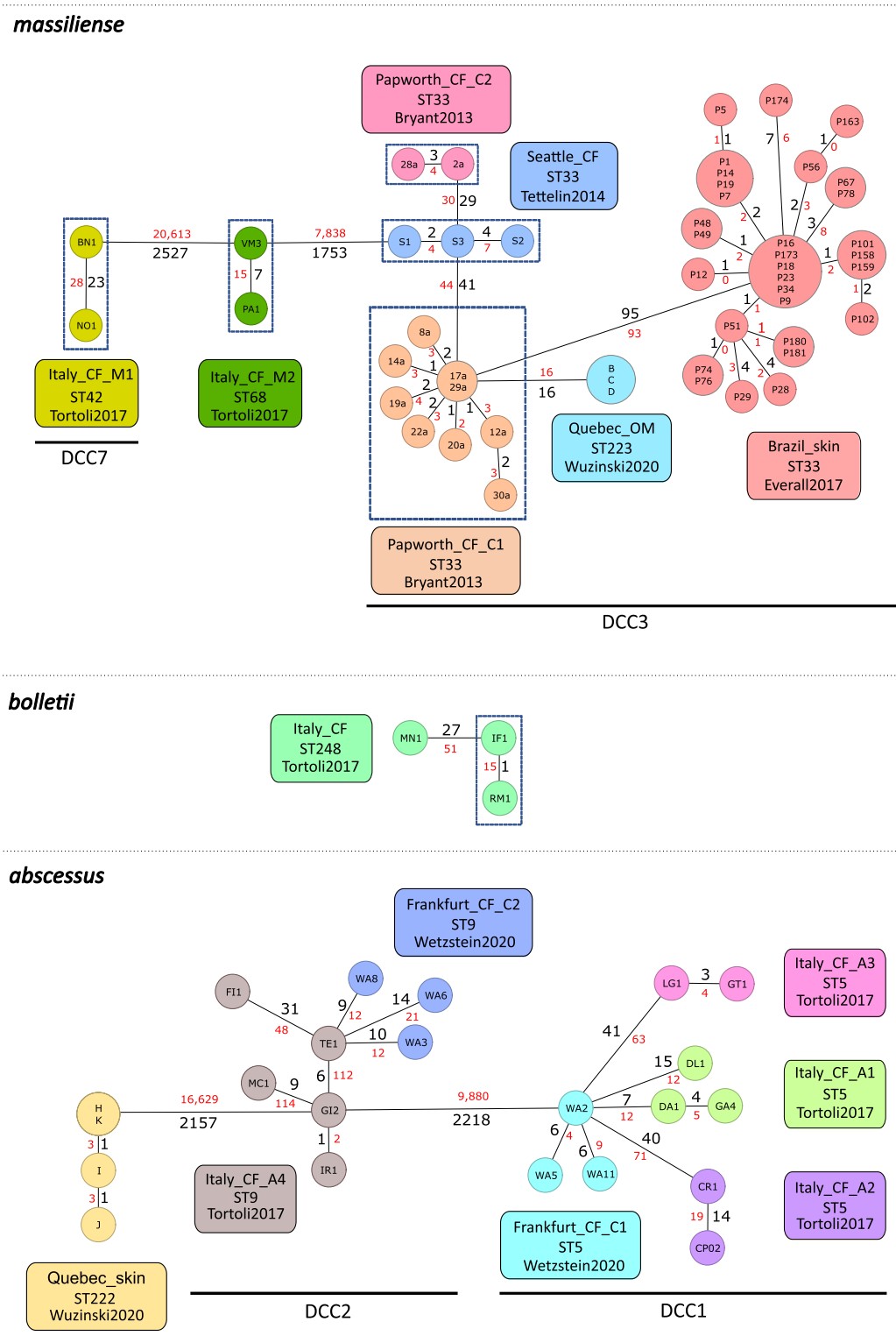

**Fig. 2 | CgMLST-based minimum spanning tree visualizing the genetic relationship between isolates from three extra-pulmonary outbreaks and 12 putative pulmonary transmission clusters.** Isolates were previously clustered in six independent studies[12, 15, 21, 41–43]. Node labels correspond to isolate ID. The size of the nodes is proportional to the number of samples with the same cgMLST profile (i.e. with allelic distance=0). Distances (SNPs in red and alleles in black) between two isolates are displayed on the branches (not to scale). Dashed boxes indicate patients that have attended the same cystic fibrosis (CF) healthcare center at the same time (i.e. putative patient-to-patient transmission event). OM: otitis media.

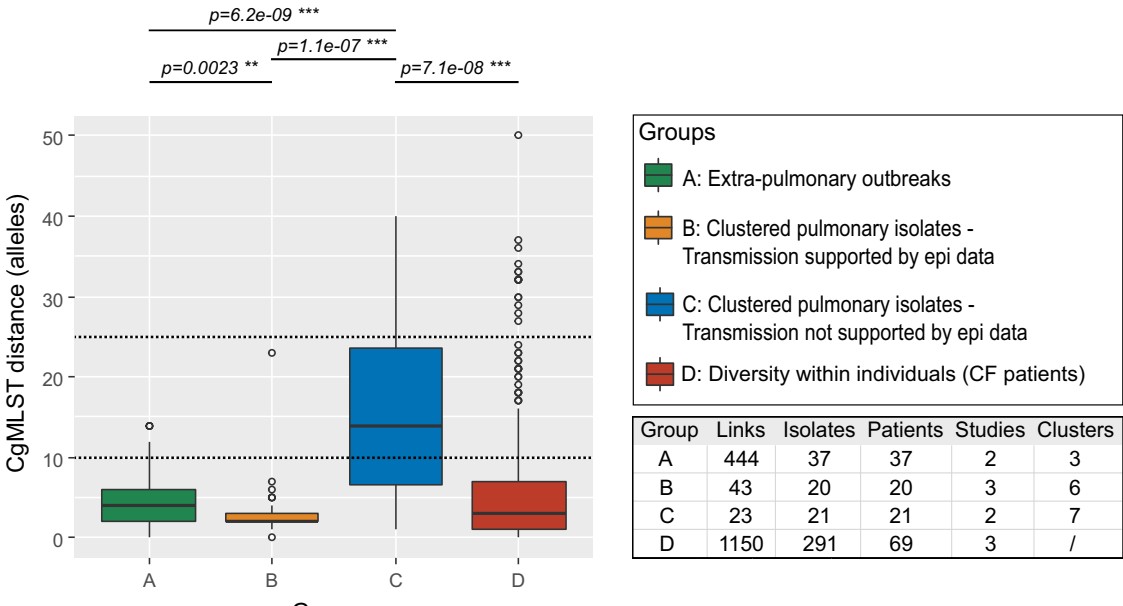

**Fig. 3 | Boxplots of pairwise genetic distances (i.e. cgMLST allele differences) between related isolates.** **A** Pairwise distances between isolates from previously reported extra-pulmonary Mab outbreaks[21, 41] (*n* = 444 pairwise comparisons). **B** Pairwise distances between pulmonary isolates from cystic fibrosis (CF) patients for which epi data suggested nosocomial transmission[12, 15, 43] (*n* = 43 pairwise comparisons). **C** Pairwise distances between pulmonary isolates from predominantly cystic fibrosis (CF) patients for which epi data did not support nosocomial transmission[15, 42] (*n* = 23 pairwise comparisons). For **B** and **C**, clusters were previously defined with a threshold of 30 SNPs. **D** Pairwise distances between isolates from the same patient[12,15,42] (*n* = 1,150 pairwise comparisons). All outliers are related to patient 5[12]. Dashed lines indicate tentative thresholds to initiate further epidemiological investigations. Solid lines indicate median pairwise allele distance, box represent interquartile range (IQR), whiskers extend to 1.5 of the IQR, dots represent outliers. Links are the amount of pairwise comparisons included per group. Pairwise allele distances between groups were compared with a Wilcoxon rank sum test (with Bonferroni correction): \**p* < 0.05, \*\**p* < 0.01, \*\*\**p* < 0.001.

## Within-patient diversity and evolution

To determine the genetic diversity of isolates from the same patient sampled at different time points (within-patient diversity), we analyzed the cgMLST profiles of 291 sequential isolates from 69 chronically infected CF patients included in three previous studies[12,15,42]. For each patient, there were between two and thirty isolates, sampled over a period of maximum 12 years (Supplementary Data 3 and 9). Except for one patient (patient 5[12]), the distance between any two isolates from the same patient did not exceed 16 alleles and 89% of same-patient isolates differed by less than 10 alleles (Fig. 3). The median number of allele differences between same-patient isolates (i.e. 3 alleles) was similar to the median pairwise distances observed among epidemiological related isolates (four and two for extra-pulmonary and pulmonary isolates, respectively) (Fig. 3).

Minimum spanning trees from sequential isolates of CF patients often showed a star-like structure consistent with the clonal evolution of a monomorphic pathogen (Fig. 4). Interestingly, some strains were genetically highly stable. This was especially notable for two patients, i.e. IF and RM[15], where the same cgMLST type was sampled over 12 years (Fig. 4). On the other hand, larger genetic distances of more than 20 alleles between same-patient isolates sampled within the same year or even same month (e.g. patient 3 and 5, Figs. 3 and 4) were also observed.

If only pairwise distances to the first available isolate were considered, the within-patient divergence was lower than 10 alleles within 12 years for 64 out of 69 (93%) of the patients (Supplementary Fig. 7). If the five outliers were removed, we observed a weak positive linear correlation of allelic distance and time with an evolutionary rate of 0.45 alleles/genome/year (95% CI 0.13–0.76 and $R^2$ = 0.105) for $Mab_A$, 0.28 alleles/genome/year (95% CI 0.09–0.46 and $R^2$ = 0.079) for $Mab_M$, and 0.44 alleles/genome/year (95% CI 0.08-0.81 and $R^2$ = 0.294) for $Mab_B$ (Supplementary Fig. 7). The initial genetic diversity (within the first year) was 1.9 alleles [1.0–2.7] for $Mab_A$, 1.6 alleles [1.2–2.0] for $Mab_M$, and 1.3 alleles [0–4.0] for $Mab_B$.

## Discussion

In this study, we developed a robust cgMLST scheme for the emerging pathogen *Mycobacterium abscessus*, which can delineate Mab population structure, outbreaks and within-patient diversity. The high discriminatory power has the ability for an early detection of infection sources, transmission hotspots, and yet undetected contact cases. The stable scheme (i.e. fixed loci) with a harmonized expandable nomenclature (i.e. allele numbers) allows direct comparability of results by different laboratories and facilitates prospective global Mab surveillance. Importantly, we also defined thresholds for cgMLST-based cluster analysis, e.g. to determine Mab outbreaks in the hospital setting, and to classify new isolates at subspecies level and within known global complexes.

Several researchers have considered a cgMLST stable if at least 95% of the cgMLST genes are present in all or most strains[44–46]. Our newly developed cgMLST Mab scheme consists of 2904 core loci of which at least 95% were found in 99.4% of a diverse set of 1797 Mab isolates. All seven dominant circulating clones (DCC) previously defined by Ruis and coworkers[26] were confirmed by cgMLST analysis of the population structure of the large global dataset investigated in our study. We also showed that isolates with unknown taxonomy could be classified as DCC using a maximum pairwise distance of 250 alleles compared to a DCC reference genome, thereby offering a convenient alternative to classification based on positioning within a large reference tree. In addition, we illustrate that the DCC nomenclature is largely congruent with distinct STs of the traditional 7-loci MLST scheme[47]. However, we propose a differentiation of DCC3 into two clades which are clearly distinguished by (i) two distinct ST types, i.e. DCC3a (associated with ST33) and DCC3b (associated with ST37) and (ii) inter-DCC pairwise distances of >250 alleles. These results underline that traditional MLST analysis is compatible with cgSNP and cgMLST analysis and can still distinguish between the global

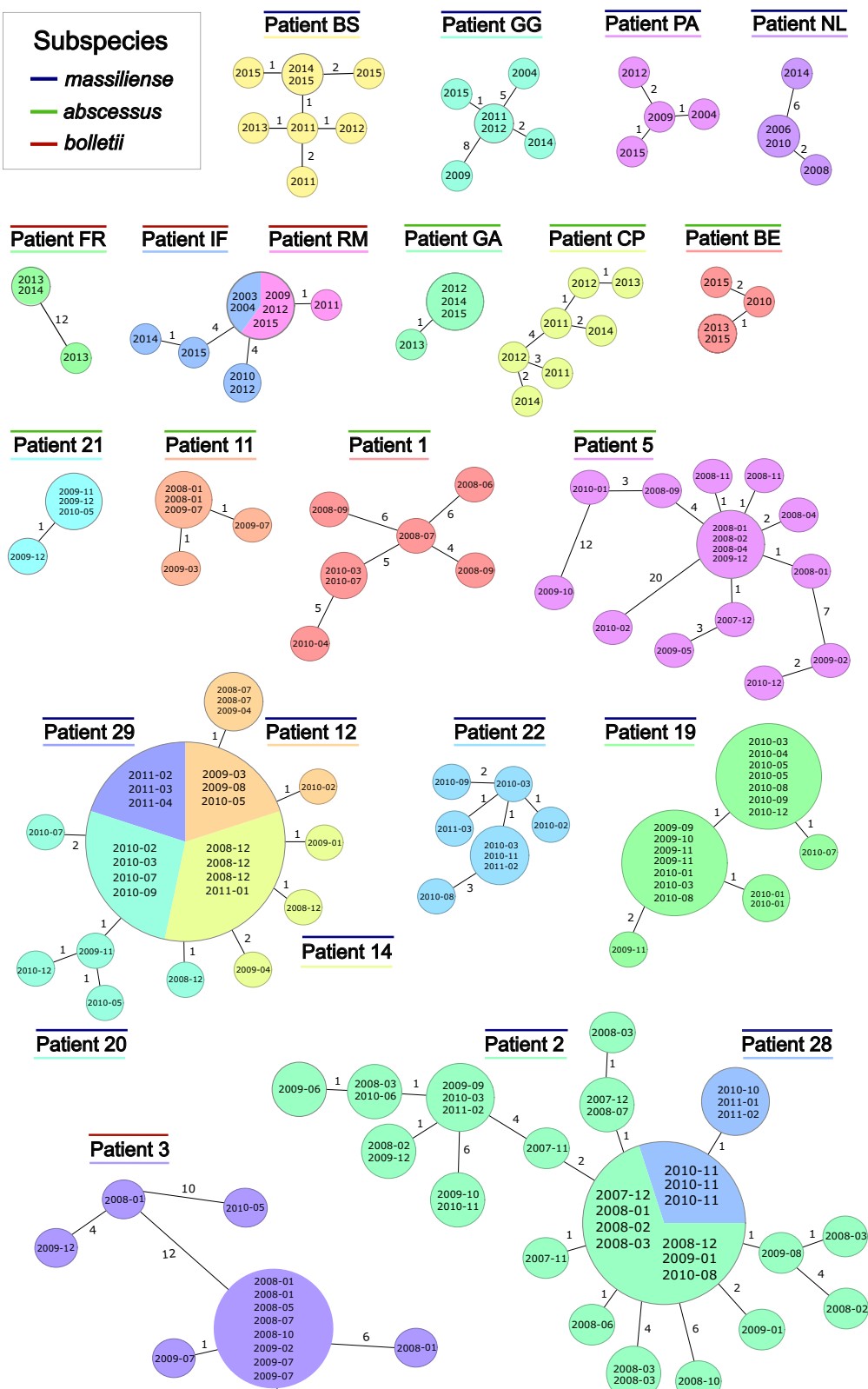

**Fig. 4 | CgMLST-based minimum spanning tree visualizing the genetic relationship between sequential isolates from the same patient (within-patient diversity).** Patients from two studies[12, 15] with more than three isolates are displayed. Node labels correspond to the date of *Mycobacterium abscessus* isolate collection. The size of the nodes is proportional to the number of samples with identical cgMLST profile (i.e. with distance=0). Allelic distances between two isolates are displayed on the branches (not to scale).

DCCs, however, it is not suited for the detection of recent transmission events or outbreaks.

Analysis of the within-patient diversity revealed that some patients were infected with a highly persistent clone that was sampled for over a decade without any allelic differences. On the other hand, we also observed considerable genetic diversity (>20 alleles) between isolates sampled within the same year for some patients. This might be the result of a mixed infection with a phylogenetically related clone or an undetected long-term chronic Mab infection with subsequent diversification of the infecting clone into different co-existing subpopulations. Also the presence of a putative "hypermutator" strain has been previously discussed[48].

Overall, 99% of pairwise distances (1620 out of 1637 comparisons) between epidemiologically linked isolates, i.e. isolates from the same patient, isolates belonging to the same extra-pulmonary outbreak or isolates from CF patients with suggested nosocomial transmission, were below 25 alleles and 90% (1484 out of 1637 comparison) below 10 alleles. On the other hand, several isolates from patients for which there was no obvious epidemiological link could also be linked with less than 25 alleles. Therefore, we propose a threshold of 25 for initial clustering and indicating "possible" transmission and a threshold of 10 for "probable" recent transmission (whether direct or indirect). This is similar to previously defined SNP thresholds for Mab[12,15,25,49], and indicates the high discriminatory power of the developed cgMLST scheme. Still, these thresholds need to be validated in further studies, but present a valid guideline to initiate further epidemiological studies and to potentially unravel new transmission routes.

Although SNP and allele distances were typically in the same range for closely related isolates, there were also two Mab$_A$ (DCC2) isolates that could be connected with less than 10 alleles, while differing by more than 100 SNPs compared to two other cluster members. This might be the result from a horizontal gene transfer (HGT)/homologues recombination event with DNA from another Mab strain or even another mycobacterial species[48,50–52]. As recombination has shaped the Mab genome more than mutations[53,54], cgSNP is likely more prone to overestimate evolutionary distances and thereby missing epidemiological linked patients. On the other hand, genetic diversity might also be underestimated by cgMLST, as it does not include mutations in intergenic regions, accessory genes or plasmids and because multiple mutations within one gene (not caused by HGT) are translated into only one allele difference[55–57]. Therefore, estimates of evolutionary rates based on alleles should also be interpreted with caution.

Compared to cgSNP, cgMLST analysis is faster, more easily expandable with new isolates and more standardized[55,58,59]. The genes comprised in the cgMLST scheme can be identified in new samples using different algorithms (i.e. from assemblies using BLAST or from reads using kmer mapping), depending on the cgMLST software used[60–63]. Although our scheme was developed, applied and validated within a commercial software suite (SeqSphere + ), loci definitions and allele nomenclature are public (https://www.cgmlst.org) and available for implementation in other, open-source software or web-based applications (e.g. BIGSdb[60]). This means that cgMLST results can be compared directly between laboratories and public health institutions worldwide, further facilitating standardization and surveillance on a global level.

In conclusion, the whole genome sequencing-based cgMLST approach is a powerful tool for high-resolution molecular epidemiological investigations of Mab strains. It facilitates standardized prospective transmission analysis e.g. for early outbreak detection and identification of potential transmission routes in a hospital setting. The cgMLST scheme that was developed in this study is publicly accessible (https://www.cgmlst.org), allowing for efficient surveillance on a global level. Therefore, we believe that it should be part of a strategy to tackle the growing public health treat of this emerging pathogen.

## Methods

### Data collection
For this study, we downloaded assemblies (fastA files) from the NCBI/RefSeq Assembly database[64,65] for 1797 isolates (Supplementary Data 1) and Illumina paired-end read datasets (fastQ files) from the sequence read archive (SRA)[66] for 372 isolates (Supplementary Data 3). The total dataset ($n = 2169$) comprised 1991 unique biosamples. An overview of all datasets used in this study is provided in Supplementary Fig. 2.

### Public assembly set ($n = 1797$)
The genome of the Mab$_A$ ATCC 19977 type strain (accession number NC_010397.1), as well as all nonanomalous and non-suppressed genome assemblies (FastA files) annotated as *Mycobacterium abscessus* that were available on May 26th, 2021 ($n = 1,810$) were downloaded from NCBI/RefSeq[64,65]. All assemblies with more than 300 contigs ($n = 13$) were removed from further analysis, resulting in a final set of 1797 genome sequences (Supplementary Data 1). We used Mashtree v.1.2.0[67] to determine pairwise mash distances[68] and to place the 1797 assembled genomes in a neighbor-joining (NJ) tree (Supplementary Fig. 1). Isolates were taxonomically classified into one of the three subspecies based on their position in the mash-based NJ tree using known taxonomic classification of their neighbors as reference (Supplementary Data 1).

### Public read set ($n = 372$)
WGS data (FastQ files) from 30 isolates included in the technical validation set (Supplementary Methods 2) and 342 isolates from the calibration set were downloaded from SRA. The calibration set includes isolates from six previously published studies concerning Mab transmission or outbreaks[12,15,21,41–43]. Only patients which were involved in outbreak or putative transmission clusters or for which multiple longitudinal samples with the same Mab subspecies were available, were considered. For the Brazilian outbreak[41], only 2 isolates per region were retained. In addition, only isolates with more than 95% good cgMLST targets (see section "Design of cgMLST scheme") and for which the length of the respective assembly did not deviated by more than 25% compared to the Mab type strain (NC_010397.1) were included. Using these criteria, the final calibration set comprised 342 isolates from 119 patients (Supplementary Data 3).

### Genome assemblies
In addition to the 30 assemblies available from RefSeq, we also calculated 180 new assemblies for the isolates from the technical validation set. These assemblies were made starting from the 30 downloaded read sets (fastQ files) with different tools, preprocessing steps and assemblers (Supplementary Data 4). All datasets were downsampled to a coverage of 100x if applicable. The assembly tool shovill v1.1.0[38] was run on a linux 5.4.0 (Ubuntu 20.04 LTS) server with Intel®Xeon® E5-2650 v4 processor @ 2.2 GHz and 48 Gb RAM using 8 threads. Default trimming in the shovill pipeline was performed using trimmomatic v0.39[69] with the following parameters "leading:3 trailing:3 minlen:30 tophred33". Prior read error correction in the shovill pipeline was performed using lighter v.1.1.2[70]. Two assemblers available in shovill were evaluated: skesa v2.4.0[39] as well as SPAdes v3.15.0[40]. In addition, assemblies were made using skesa v2.3.0 in SeqSphere + (v7.7.5) on a windows 10 laptop with intel®Core® i7-10510U processor @ 1.8 GHz, 16 Gb RAM and 8 threads. Default trimming in SeqSphere+ includes trimming reads at 5' and 3' until average quality is ≥30 in a window of 20 bases. Mapping was performed in SeqSphere+ against the seed genome (NC_010397.1) using BWA-MEM v0.7.15.

For isolates included in the calibration set (Supplementary Data 3), assemblies were made with skesa v2.4.0 without read error correction using shovill v1.1.0.

## Design and application of cgMLST scheme

Detailed information regarding scheme creation can be found in Supplementary Methods 1 and Supplementary Fig. 2. In summary, core loci were defined using the cgMLST target definer v1.5 implemented in the SeqSphere+ software (client v7.7.5). The finished genome of the Mab$_A$ ATCC19977 type strain was used as seed genome. For the penetration set, all genome assemblies up to chromosome or complete genome level were included as well as 31 draft genomes to better represent Mab diversity (Supplementary Data 1 and Supplementary Fig. 1). All publicly available plasmids ($n = 17$ on July 2nd, 2021) were used to exclude plasmid-borne sequences from the scheme.

Extraction of cgMLST loci from genome assemblies and assignment of allele numbers was also performed in SeqSphere+. According to the default requirements of the target quality control of SeqSphere+, "good quality" cgMLST targets were defined as loci with (i) the same length as reference genes +/− 3 triplets, (ii) no ambiguities (e.g. N), (iii) no frameshifts compared to reference genes, (iv) at least 90% identity to reference sequence and (v) valid start and stop codons and no internal stop codons.

Pairwise distances between two isolates were calculated as the amount of cgMLST loci with a different allele number, ignoring missing (bad quality or absent) cgMLST loci.

## Multilocus sequence typing

The seven loci (*argH*, *cya*, *gnd*, *murC*, *pta*, *purH* and *rpoB*) included in the recently updated Mab scheme from pubMLST[47] were extracted from whole genome assemblies using BioNumerics v7.6. Unknown sequences for which no allele number was retrieved and unknown STs were submitted to the pubMLST database and subsequently were assigned new numbers.

## Core genome single nucleotide polymorphism (cgSNP) analysis

Whole genome sequence reads (FastQ files) of isolates from the outbreak/transmission set (Supplementary Data 3) were processed by the MTBseq pipeline (v.1.0.4)[31,71] with default settings using the Mab$_A$ ATCC19977 type strain genome (NC_010397.1) as a reference. The resulting core SNP alignment (80.651 variant positions) was used to calculate a pairwise distance matrix.

## Statistics

Pairwise allele distances between groups were compared with a Wilcoxon rank sum test (with Bonferroni correction) in R v.4.0.2, as we did not assume a normal distribution ($P < 0.05$, Shapiro-Wilk normality test). To explore a possible temporal signal (i.e., the rate of allele changes per genome per year) in sequential patient isolates we employed a linear regression (least square approach) of the allelic distance between the first available and a subsequent isolate against time (years) and reported $R^2$ values. The rate of change was inferred from the slope of the linear regression equation, and the initial genetic diversity (i.e., distance to the first isolate within the first year) was inferred from the intercept.

## Reporting summary

Further information on research design is available in the Nature Research Reporting summary linked to this article.

## Data availability

Accession numbers of all whole genome sequencing datasets analyzed in this study are listed in Supplementary Data 1 and 3. The cgMLST scheme is publicly available at cgmlst.org (https://www.cgmlst.org/ncs/schema/22602285/).

## Code availability

Custom scripts used for this study and a detailed manual to perform cgMLST analysis for *M. abscessus* isolates within the commercial software SeqSphere+[37] can be found at github (ngs-fzb/NTMtools)[72].

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

## Acknowledgements

We thank Dag Harmsen and Jörg Rothgänger for their technical help with regard to the construction of the cgMLST scheme. This work was supported by a financial grant from Mukoviszidose Institut gGmbH, Bonn, the research and development arm of the German Cystic Fibrosis Association Mukoviszidose e.V (project number 2004 – FM). Parts of this work have been supported by the Deutsche Forschungsgemeinschaft (DFG, German Research Foundation) under Germanys Excellence Strategy – EXC 2167 Precision Medicine in Inflammation, the German Ministry of Education and Research (BMBF) for the German Center of Infection Research (DZIF), and the Leibniz Science Campus Evolutionary Medicine of the LUNG (EvoLUNG).

## Author contributions

F.M., S.N., and M.D. conceived and directed the project. M.D. and M.M. wrote the first draft of the manuscript. N.W., T.K. and all other authors commented and edited various versions of the draft manuscript. All authors read and approved the final manuscript.

## Funding

## Competing interests

The authors declare no competing interests.
