## [Peer Review File · Nature Communications]

Delineating *Mycobacterium abscessus* population structure and transmission employing high-resolution core genome multilocus sequence typingREVIEWER COMMENTS

Reviewer #1 (Remarks to the Author):

Mago Diricks and co-workers report a rapid, replicable, superior, and easy typing scheme for epidemiological investigation of *Mycobacterium abscessus* (Mab) which is an emerging and multidrug-resistant pathogen causing severe respiratory, skin, and mucosal infections. The genomic data set used by the authors for technical validation and calibration is already available in NCBI databases and published with all the metadata. The scheme is based on a well-defined core gene set (cgMLST) of Mab. The authors have provided evidence that the scheme is more powerful than the existing methods based on genome-wide SNPs in understanding population structure, dynamics, and transmission at different levels. They also provide evidence that the cgMLST is rapid, easy, and robust and provide high-resolution efficient surveillance of the pathogen on a global level. The method is also promising for early detection and contact tracing. They define thresholds and provide evidence at the taxono-phylogenetic level, dominant circulating clones/clusters, hospital-level both between patients and also outbreaks apart from within-patient diversity and evolution.

Below are major comments that need to be addressed by the authors

1. In the abstract, the authors mention using more than 2000 genomes which needs to be corrected as the number is less than 2000...
2. It is not clear regarding the set of 97 genomes, particularly 30 draft genomes used for initial validation...from a global point of view and also a diversity point of view..more details need to be provided and in the main text.
3. In cases where there is incongruence with cgMLST and cgSNP, details are lacking...for example in two isolates from patients GI2 and MC1/TE1 (line 156), details of the 23 consecutive genes.. what is annotation status and conservation in Mabs?
4. In continuation of the above comment, the paragraph from lines 261 to 266 is vague...like exactly how many isolates with less than 10 alleles while differing by more than 100 SNPs...what are the isolates, metadata correlation, evidence for the HGT, recombination or even mutations in genes implicated in mutator phenotype.
5. With respect to the analysis of the global population structure, why the strains with less than 95% good cgMLST targets are in MabM (8 out of 563) compared to MabA (just 2 out of 1110). Is cgMLST more efficient in MabA compared to MabM... Focusing on results and discussion in this context is required.

Reviewer #2 (Remarks to the Author):

This is a clearly written manuscript and the cgMLST scheme the authors have proposed will be useful to the community, as a systematic way of identifying the globally distributed dominant clones of *M. abscessus* is needed. The methodology and analysis appears to be sound.

Major comments:

-Using this scheme will be of interest to people wanting to identify the subspecies or DCC status of newly sequenced isolates. However the authors haven't provided a clear analytical pathway to how this might be achieved. It would be useful if the authors could summarise the steps to how these genes might be identified (blast/mapping?) from their data.

-This scheme is only useful if it is made available, and can be compared easily. However I can't find the *M. abscessus* scheme on <https://cgmlst.org/ncs> so it is hard for me to assess this

Minor comments:

-please be more precise when stating this analysis is "fast". How does it compare with standard reference based mapping and identification of DCCs via phylogenetic position?

-please reference or describe the "MTBseq pipeline" on line 305

Please note that all changes to the manuscript are showed with track changes.

Please note that we needed to change the order of the supplementary figures and tables, as new figures and tables were provided.

Reviewer #1 (Remarks to the Author):

Margo Diricks and co-workers report a rapid, replicable, superior, and easy typing scheme for epidemiological investigation of *Mycobacterium abscessus* (Mab) which is an emerging and multidrug-resistant pathogen causing severe respiratory, skin, and mucosal infections. The genomic data set used by the authors for technical validation and calibration is already available in NCBI databases and published with all the metadata. The scheme is based on a well-defined core gene set (cgMLST) of Mab. The authors have provided evidence that the scheme is more powerful than the existing methods based on genome-wide SNPs in understanding population structure, dynamics, and transmission at different levels. They also provide evidence that the cgMLST is rapid, easy, and robust and provide high-resolution efficient surveillance of the pathogen on a global level. The method is also promising for early detection and contact tracing. They define thresholds and provide evidence at the taxono-phylogenetic level, dominant circulating clones/clusters, hospital-level both between patients and also outbreaks apart from within-patient diversity and evolution.

Below are major comments that need to be addressed by the authors

1. In the abstract, the authors mention using more than 2000 genomes which needs to be corrected as the number is less than 2000...

Thank you for pointing this out and we apologize for the confusion regarding the datasets used. We did in fact analyse over 2000 assemblies, but we agree that these corresponded to only 1991 unique isolates (i.e. unique BioSamples). This results from the fact that for some isolates, different assemblies were analysed for technical and calibration purposes. All unique Biosample Accession numbers are now listed in **Supplementary Table 1 and Table 3** for clarification. We also adjusted **Supplementary Fig. 2** for clarification.

In addition, we changed the number of unique datasets used in this manuscript in the abstract as follows "*Whole genome sequences of 1991 isolates...*".

We further clarify these numbers in the method section as follows:

Data collection

For this study, we downloaded assemblies (fastA files) from the NCBI/RefSeq Assembly database for 1,797 isolates (Supplementary Table 1) and Illumina paired-end read datasets (fastQ files) from the sequence read archive (SRA) for 372 isolates (Supplementary Table 3). The total data set (n=2,169) comprised 1991 unique biosamples. An overview of all datasets used in this study is provided in Supplementary Fig. 2.

Public assembly set (n=1,797)

...

Public read set (n=372)

WGS data (FastQ files) from 30 isolates included in the technical validation set (see Supplementary Information section technical validation of cgMLST scheme) and 342 isolates from the calibration set were downloaded from SRA.

...

Genome assemblies

In addition to the 30 assemblies available from RefSeq, we also calculated 150 new assemblies for the isolates from the technical validation set. These assemblies were made starting from the 30 downloaded read sets (fastQ files) with different tools, preprocessing steps and assemblers (Supplementary Table 4).

2. It is not clear regarding the set of 97 genomes, particularly 30 draft genomes used for initial validation...from a global point of view and also a diversity point of view..more details need to be provided and in the main text.

We added additional information about the selection of the 97 genomes for scheme creation (scheme creation set) in the main text, section Results - Design and technical validation of a stable Mab cgMLST scheme:

“We used MabA type strain ATCC19977 and 96 additional publicly available assemblies (scheme creation set) from a genetically diverse and global set of Mab isolates to define a hard core genome with SeqSphere+ software (Supplementary Table 1, Supplementary Fig. 1-3 and Supplementary Information section creation of cgMLST scheme). The scheme creation set included representatives for all subspecies, for seven previously defined DCCs26 as well as non-DCC strains, which were collected in at least 11 different countries. The resulting cgMLST scheme consists of 2,904 loci (Supplementary Table 2), representing 59% of the gene set from MabA type strain ATCC19977.”

and in section Methods - Design and application of cgMLST scheme

„ Detailed information regarding scheme creation can be found in Supplementary Information section creation of cgMLST sheme and Supplementary Fig. 2. In summary, core loci were defined using the cgMLST target definer v1.5 implemented in the SeqSphere+ software (client v7.7.5). The finished genome of the MabA ATCC19977 type strain was used as seed genome. For the penetration set, all genome assemblies up to chromosome or complete genome level were included as well as 31 draft genomes to better represent Mab diversity (Supplementary Table 1 and Supplementary Fig. 1).”

In addition, we provided more details regarding scheme creation and technical validation in Supplementary Information (see tracked changes).

A visual representation of the genomes within the full phylogeny is provided in **Supplementary Fig. 1**, which we updated to better understand the selection of the additional draft genomes (included in the scheme creation set) and the difference between scheme creation and technical validation set. In addition, we updated **Supplementary Fig. 2 and 3** to make the differences between the datasets more clear.

3. In cases where there is incongruence with cgMLST and cgSNP, details are lacking...for example in two isolates from patients GI2 and MC1/TE1 (line 156), details of the 23 consecutive genes.. what is annotation status and conservation in Mabs?

4. In continuation of the above comment, the paragraph from lines 261 to 266 is vague...like exactly how many isolates with less than 10 alleles while differing by more than 100 SNPs...what are the isolates, metadata correlation, evidence for the HGT, recombination or even mutations in genes implicated in mutator phenotype.

We thank the reviewer for pointing towards this interesting observation. First, we would like to note that there were only 20 consecutive genes involved instead of 23 (this was a typing mistake). We have provided an additional Table (**Supplementary Table 8**) with the annotation status of these genes and the number of SNPs (i.e. > 100 SNPs) between the two isolates from the Italian cluster (IR1 and GI2) and the two other cluster members (MC1 and TE1). Each of these four isolates were collected from different patients in Italy. This Table also shows that eight out of 20 consecutive genes were not included in the cgMLST scheme, most of them because they were not conserved in all genomes from the scheme creation set.

If there is no contamination (e.g. mixed sample) or assembly artefact, a high density of SNPs within a larger region is typically a good indication for recombination/HGT². All four isolates had >95% good cgMLST targets and the assembly lengths of GI2 (5.4 Mbp and 104 contigs) and IR1 (5.3 Mbp and 109 contigs) were only slightly larger compared to Mab type strain ATCC19977 (5.1 Mbp). In addition, the contigs of GI2 and IR1 with the high-density SNP region also harbored adjacent genes without SNPs, and a reference mapping approach with MTBseq³ did not show an unusual coverage across the 20 genes. Lastly, we found that the “high-density SNP region” affecting 20 consecutive genes in the GI2 and IR1 isolates had a higher blastN score for *M. immunogenum* compared to *M. abscessus*, which might point towards a recombination event with another mycobacterial species. Thus, we can only speculate that this rare accumulation of SNPs in few consecutive genes might be the result of a homologous recombination event rather than a mixed infection/contamination. We would like to perform additional analyses to further scrutinize this phenomena, but we consider this out of scope for this publication.

Therefore, we amended the results as follows:

“For most closely related isolates, pairwise SNP and allele distances were in the same range (Figure 2). However, for two MabA isolates from cluster Italy_CF_A4 (GI2 and IR1), the pairwise SNP distance compared to the other cluster members MC1 and TE1 was much higher (>100 SNPs) compared to the number of allelic differences (<10 alleles). More detailed analysis revealed that the majority of these SNPs in the GI2 and IR1 genome were concentrated in 20 consecutive genes (MAB_1023c-1042c), eight of which were not included in the cgMLST scheme (Supplementary Table 8). BLASTN search with this high-density SNP region revealed a higher total score for Mycobacterium immunogenum (25,261 with 95% coverage and 87.21% identity) compared to the top Mab hit (24,515, 89% coverage and 87.37% identity), further pointing towards a putative recombination event.”

We amended the discussion as follows:

“Although SNP and allele distances were typically in the same range for closely related isolates, there were also two MabA (DCC2) isolates that could be connected with less than 10 alleles, while differing by more than 100 SNPs compared to two other cluster members. This might be the result from a horizontal gene transfer (HGT)/homologues recombination event with another Mab strain or even another mycobacterial species.

5. With respect to the analysis of the global population structure, why the strains with less than 95% good cgMLST targets are in MabM (8 out of 563) compared to MabA (just 2 out of 1110). Is cgMLST

more efficient in MabA compared to MabM... Focusing on results and discussion in this context is required.

Although the percentage of good cgMLST targets is slightly higher for MabA (99.8%) compared to MabM (98.6%), we don't believe this is problematic for genotyping of MabM. In the end, 7 out of 8 MabM isolates that had <95% good cgMLST targets, still had >90% good cgMLST targets, which is in SeqSphere+ by default still considered as acceptable for comparative analysis. In addition, we noticed that the MabM isolates were scattered across the tree and not e.g. concentrated in a clade that is more distantly related to subsp. *abscessus* (we have updated **Supplementary Fig. 1** to visualize this), indicating that the "Low percentage" might also have to do with the quality of the sequencing data itself and not necessarily with the scheme.

Nevertheless, we used a conservative approach and decided to remove all isolates with <95% cgMLST targets from further analysis in the main manuscript to not disturb the global phylogeny.

We have amended the results as follow:

„For 1,786 out of 1,797 (99.4%) datasets, more than 95.0% good cgMLST targets were found and for 1,796 (99.9%) more than 90% of the cgMLST genes were present, indicating a stable core genome applicable for all Mab strains“.

We have amended the discussion as follow:

„Several researchers have considered a cgMLST stable if at least 95% of the cgMLST genes are present in all or most strains⁴⁻⁶. Our newly developed cgMLST Mab scheme consists of 2,904 core loci of which at least 95% were found in 99.4% of a diverse set of 1,797 Mab isolates.“

Reviewer #2 (Remarks to the Author):

This is a clearly written manuscript and the cgMLST scheme the authors have proposed will be useful to the community, as a systematic way of identifying the globally distributed dominant clones of *M. abscessus* is needed. The methodology and analysis appears to be sound.

Major

comments:

-Using this scheme will be of interest to people wanting to identify the subspecies or DCC status of newly sequenced isolates. However the authors haven't provided a clear analytical pathway to how this might be achieved. It would be useful if the authors could summarise the steps to how these genes might be identified (blast/mapping?) from their data.

Thank you very much for pointing this out. This motivated us to propose a reference set of 10 isolates that can be used to classify newly sequenced isolates into one of three subspecies and into DCCs using cgMLST analysis. These isolates were part of the public assembly set (Supplementary Table 1) but we summarized them in a new supplementary Table : **Supplementary Table 5** for clarity. To be able to classify newly sequenced isolates into DCCs solely based on one representative of each DCC, we performed an additional analysis with the public assembly set and established a new threshold, which is visualized in a new figure (**Supplementary Fig. 6**). Additionally, we wrote a detailed manual to perform cgMLST analysis in SeqSphere+ and to identify subspecies/DCC classification of unknown isolates, which we made available at <https://github.com/ngs-fzb/NTMtools>.

We hope that with these efforts, the use of the scheme will be facilitated.

In addition, we added to the results – section Analysis of global populations structure and discussion:

“Consistent with this finding, all isolates belonging to MabA, MabM and MabB were most closely related (i.e. had the lowest amount of allele differences) to the type strains for subsp. abscessus (strain ATCC19977; accession NC_010397.1), massiliense (JCM 15300; NZ_AP014547.1) and bolletii (BD; NZ_AP018436.1), respectively. These three type strains (Supplementary Table 5) can thus be used for distance-based classification of new isolates at subspecies level, without the need for phylogenetic tree building”.

And

“Next, isolates with unknown DCC status that were positioned within a clade in the NJ tree containing isolates with known DCC status (Figure 1) were classified into the corresponding DCC (Supplementary Table 1). Using a set of eight representatives, one for each DCC (including 3a and 3b; Supplementary Table 5), we found a clear separation between intra-DCC pairwise distances (i.e. distances between the representative of the DCC and isolates belonging to the same DCC) and pairwise distances between the representative and isolates not belonging to the corresponding DCC (Supplementary Fig. 6). More concrete, the majority of DCC isolates had less than 250 allele differences compared to the corresponding reference genome (Supplementary Fig. 6).”

We amended the discussion section as follows:

Compared to cgSNP, cgMLST analysis is faster, more easily expandable with new isolates and more standardized. The genes/alleles comprised in the cgMLST scheme can be identified in new samples using different algorithms (i.e. from assemblies using BLAST or from reads using kmer mapping), depending on the cgMLST software used. Although our scheme was developed, applied and validated within a commercial software suite (SeqSphere+), loci definitions and allele nomenclature are public (<https://www.cgmlst.org>) and available for implementation in other, open-source software or web-based applications (e.g. BIGSdb⁷).

We amended the abstract as follows:

“We confirmed and amended the nomenclature of the main dominant circulating clones (DCC) and found that these also correlate well with traditional 7-loci MLST. DCC strains could be linked to a corresponding DCC reference genome with less than 250 alleles while 99% of pairwise comparisons between epidemiologically linked isolates were below 25 alleles and 90% below 10 alleles.”

-This scheme is only useful if it is made available, and can be compared easily. However I can't find the M. abscessus scheme on <https://cgmlst.org/ncs> so it is hard for me to assess this

We apologize for this inconvenience. The scheme is now publicly available at <https://www.cgmlst.org/ncs/schema/22602285/>. Gene and allele definitions can be downloaded from here. Please let us know if you still experience problems to access it.

Minor comments:

-please be more precise when stating this analysis is "fast". How does it compare with standard reference based mapping and identification of DCCs via phylogenetic position?

Details about calculation time for cgMLST analysis are provided in Supplementary Table 4 and discussed in Supplementary Information section technical validation of cgMLST scheme. As the calculation time for SNP analysis is highly dependent on the pipeline used, the amount of isolates

included and the hardware configuration used, we would like to refrain from providing exact numbers for SNP analysis in this manuscript. However, cgMLST analysis is inherently faster as only 2000~3000 allele numbers (in our case 2904 loci) need to be compared as opposed to millions of bases for SNP analysis (also mentioned by Eyre and coworkers⁸). Using our custom cgSNP pipeline ³, comparative analysis for a set of 30 isolates takes about 2.3 hours. If the dataset needs to be expanded with one isolate, it would again take 2.3 hours and a lot more if a bigger set needs to be analysed. With cgMLST analysis, on the other hand, once the cgMLST profiles of all isolates are calculated (i.e. the 2904 allele numbers), comparing the profiles of 30 or even 1000 of isolates only takes a couple of minutes.

Without wanting to go in detail, we amended the discussion section as follows:

“Compared to cgSNP, cgMLST analysis is faster, more easily expandable with new isolates and more standardized⁸⁻¹⁰.”

-please reference or describe the "MTBseq pipeline" on line 305

Thank you very much for noting that this was not referenced. We added the appropriate reference accordingly.

References

1. Ruis, C. *et al.* Dissemination of *Mycobacterium abscessus* via global transmission networks. *Nat. Microbiol.* 2021 610 **6**, 1279–1288 (2021).
2. Croucher, N. J. *et al.* Rapid phylogenetic analysis of large samples of recombinant bacterial whole genome sequences using Gubbins. *Nucleic Acids Res.* **43**, e15–e15 (2015).
3. Kohl, T. A. *et al.* MTBseq: A comprehensive pipeline for whole genome sequence analysis of *Mycobacterium tuberculosis* complex isolates. *PeerJ* **2018**, e5895 (2018).
4. Neumann, B. *et al.* A Core Genome Multilocus Sequence Typing Scheme for *Enterococcus faecalis*. *J. Clin. Microbiol.* **57**, (2019).
5. Ruppitsch, W. *et al.* Defining and evaluating a core genome multilocus sequence typing scheme for whole-genome sequence-based typing of *Listeria monocytogenes*. *J. Clin. Microbiol.* **53**, 2869–2876 (2015).
6. Ghanem, M. & El-Gazzar, M. Development of *Mycoplasma synoviae* (MS) core genome multilocus sequence typing (cgMLST) scheme. *Vet. Microbiol.* **218**, 84–89 (2018).
7. Jolley, K. A. & Maiden, M. C. J. BIGSdb: Scalable analysis of bacterial genome variation at the population level. *BMC Bioinformatics* **11**, 595 (2010).
8. Eyre, D. W., Peto, T. E. A., Crook, D. W., Sarah Walker, A. & Wilcox, M. H. Hash-Based Core Genome Multilocus Sequence Typing for *Clostridium difficile*. *J. Clin. Microbiol.* **58**, (2019).
9. Uelze, L. *et al.* Typing methods based on whole genome sequencing data. *One Heal. Outlook* 2020 21 **2**, 1–19 (2020).
10. Janezic, S. & Rupnik, M. Development and Implementation of Whole Genome Sequencing-Based Typing Schemes for *Clostridioides difficile*. *Front. Public Heal.* **7**, (2019).

REVIEWERS' COMMENTS

Reviewer #1 (Remarks to the Author):

The comments are adequately addressed in the revised version.

Reviewer #2 (Remarks to the Author):

The authors have addressed all my comments. Thankyou